# Benefits Perceived by Caregivers of Patients with Alzheimer’s Disease on Physical and Emotional Health in an Interdisciplinary Program: A Qualitative Study

**DOI:** 10.3390/healthcare12141414

**Published:** 2024-07-16

**Authors:** Javier Urbano-Mairena, Javier De Los Ríos-Calonge, Salvador Postigo-Mota, Julián Carvajal-Gil, Elisa Sofía Silveira-Saraiva, Joan Guerra-Bustamante, Laura Muñoz-Bermejo

**Affiliations:** 1Social Impact and Innovation in Health (InHEALTH) Research Group, University Centre of Mérida, University of Extremadura, 06800 Mérida, Spain; carvajal@unex.es; 2Department of Sport Sciences, Sports Research Centre, Miguel Hernández University of Elche, 03202 Elche, Spain; javier.rios01@goumh.umh.es; 3Farmacogenética de Enfermedades Psiquiátricas, Faculty of Medicine, University of Extremadura, 06006 Badajoz, Spain; spostigo@unex.es; 4Queercus Software Engineering Group, Escuela Politécnica, University of Extremadura, 10003 Cáceres, Spain; sofiasilveira@unex.es; 5Physical and Health Literacy and Health-Related Quality of Life (PHYQOL) Research Group, Faculty of Sport Sciences, University of Extremadura, 10003 Cáceres, Spain; joangb@unex.es

**Keywords:** Alzheimer’s disease, qualitative study, informal caregivers, psychoemotional factors

## Abstract

Alzheimer’s disease is the most common type of dementia, severely affecting the families and caregivers who live with those affected. The aim was to explore the physical, psychological, and behavioral benefits for caregivers of people with Alzheimer’s disease. Fifteen semi-structured interviews were conducted with informal Alzheimer’s caregivers upon completion of the program. Following a discussion on the topics, categories, and codes among the researchers, a consensus was reached to obtain the final themes and categories. Three main categories were obtained: (1) perceived benefits by the participants; (2) applicability of the knowledge; and (3) proposals for improvement. Participants expressed having perceived improvements in their ability to manage emotions and cope with the situation created by the disease, physical capacity, and in their relationships. In this sense, the application of the contents addressed during the intervention became a fundamental tool for the participants’ daily lives. This study showed how an interdisciplinary intervention with psychological sessions, health education, and physical activities could be beneficial for improving both the physical and mental health of caregivers.

## 1. Introduction

Alzheimer’s disease (AD), the most common type of dementia, is a degenerative disease characterized by memory loss, the loss of cognitive function, and functional impairment, with associated neuropsychological symptoms [1,2].

It is estimated that there are around 47 million people with dementia worldwide, and according to population projections, this number is estimated to double every 20 years reaching 131.5 million people by 2050 [3].

In addition, more than 11 million family and other unpaid caregivers were providing approximately 18 billion hours of care to people with Alzheimer’s or other dementias in 2022 [4].

Family and informal caregivers living with people with dementia are profoundly affected as they provide high-intensity care that involves many hours per day, requires significant effort, often for years [5].

The widespread involvement of activities of daily living (ADLs) that dementia processes entail involves many tasks, time, and dedication on the part of the caregiver. Communication between the person with dementia and their caregiver is also increasingly impaired, making understanding each other and being understood an extremely difficult task. This ‘care time’ that the caregiver invests in the person with dementia reduces the time for themselves and has been linked to caregiver stress levels [6], role overload, and lack of information, along with financial problems and changes in the caregiver’s health status. All this leads to a decrease in quality of life.

Interventions to reduce the stressors affecting caregivers are varied and include individual, family, or group therapies. Psycho-educational programs [7,8] have been developed and found to significantly improve knowledge of the disease and coping with it and decrease anxiety levels. Cognitive–behavioral and educational programs [9,10] have shown a decrease in subjective caregiver burden and an improvement in family functioning and caregiver stress levels. In addition, multi-component programs targeting caregiver depression, burden, behaviors and self-care, social support, and behavioral and psychological symptoms [11,12,13,14,15] have also been developed with positive results on caregiver quality of life.

Although it has been shown in different studies that carers often do not have the time to participate in preventive health activities, such as regular physical training [16,17], we know that physical exercise has direct psychological benefits and can reduce subjective burden [18,19,20]. In this sense, didactic exercise interventions have been developed for people with AD and their caregivers [21,22].

So far, several studies have evaluated from a qualitative perspective the process of adaption to the disease by family caregivers and the positive and negative experiences of caregiving [23,24]. However, no study has evaluated the perceived benefits for caregivers of people with AD after participating in a interdisciplinary program with psychoemotional education, care and social support, and physical activity training interventions. A qualitative methodology in longitudinal studies could have important benefits in the lives of participants [25]. Approaching perception using qualitative methods can provide a more complete picture of program benefits in this population as psychosocial aspects, which are not easily measured, expressed, and interpreted from a quantitative perspective, help to gain insight and understanding on more complex emotional responses that quantitative methodology cannot reveal [26]. For this reason, a qualitative methodology is essential to provide a comprehensive understanding of the caregiver’s perspective.

Therefore, the aim of this study was to explore the physical and emotional benefits experienced by caregivers of people with Alzheimer’s disease, as well as to explore the knowledge acquired about the disease and its care in caregivers who participated in an interdisciplinary group program based on physical exercise sessions and emotional and care and social support in Alzheimer’s disease.

## 2. Materials and Methods

### 2.1. Study Design and Procedure

A qualitative research method was used to carry out this study, collecting data through semi-structured interviews and analyzing it through thematic analysis [27]. In order to provide a more complete picture of the benefits and perceptions of caregivers following the interdisciplinary intervention, this study was approached using qualitative methods. Among the positive aspects of this methodology is a better understanding of the emotions of caregivers.

To ensure research rigor, the criteria established by Lincoln and Guba [28] were followed. Appendix E Table A5 shows the strategies followed in this research for each of the established criteria.

### 2.2. Intervention

The intervention was carried out in Extremadura, between October and June 2023, for 9 months, with two weekly sessions of 60 min in person; in addition, voluntarily, participants could access an informal virtual session through a private web page. These virtual sessions were known as “information pills”, and participants were able to view content covered during the sessions in a virtual and summarized form. In order for participants to be able to view each session at any time, the sessions were conducted asynchronously. Table 1 shows a total of 60 face-to-face sessions were held, 20 in each of the areas of physical activity, health education, and psychoemotional education. Each week, the sessions were conducted in a single area, with participants addressing each of the areas every 3 weeks. Appendix A, Appendix B and Appendix C
Table A1, Table A2 and Table A3.

The participants had to meet the following inclusion criteria to participate in the study: (i) serve as the primary informal caregiver of a person with AD for more than 20 h per week for at least 3 months and intend to continue for the next 12 months; (ii) not have any pathologies; (iii) provide the signed informed consent for the study.

### 2.3. Sample and Recruitment

Participants in the interdisciplinary program (*n* = 15) were recruited from Alzheimer’s family associations. Participants were recruited by convenience sampling. All participants enlisted for the study consented to participate and satisfied the inclusion criteria. They did not receive any type of benefits for participating in the program.

For the recruitment of participants, the Alzheimer’s family association was contacted in the first instance, where the intervention was carried out by first canvassing potential participants. After an initial survey by the association, the researchers held an information meeting on the intervention project. Once the intervention program had been drawn up, the participants were informed about the interviews. For the interviews, a date and time were set for each interview with each participant by text message, at least one week in advance, with a reminder message being sent the days before to avoid the absence of the interviewee at the interview. All interviews were conducted in the same centers where the intervention was undertaken.

### 2.4. Interviews

A total of 15 semi-structured interviews were conducted. At the end of the intervention, a semi-structured interview was conducted with each of the participants to obtain in-depth information about the experience of the participants in the interdisciplinary Integral Care program. Each interview lasted between 15 and 35 min. Only the interviewer and the participant were in the room during the interview. The interviews were audio-recorded with the prior recorded consent of the interviewed participant. Given the private and personal nature of the information revealed during the recording, the anonymity of the participants was protected through the use of numbers or codes.

The interview was of a semi-structured narrative nature, seeking to inquire about aspects such as the development of social skills; approach to personal situations; emotional changes; care techniques for the patient; benefits perceived by the participants; and general and specific aspects of the intervention program. In addition, data were collected on age, gender, degree of relationship to the patient, sufficient economic income, level of education, and marital status. Clarifications on the questions were made when necessary. No interviews were repeated. The semi-structured interview is shown in Appendix D Table A4.

### 2.5. Analysis

All recorded interviews were transcribed by researchers J.U.M. and J.R.C. in the language in which they were conducted. In the writing of the article, selected text fragments from the transcribed interviews were translated into English.

The transcribed interviews were analyzed following a thematic analysis, which is used to recognize patterns of meaning and to make sense of the qualitative data [27].

This was followed by a first reflexive reading of all interviews, developing a logical perspective of the interviews. Subsequently, the original transcribed text of the interviews was analyzed by researcher J.U.M. and Ph.D. L.M.B. The MAXQDA 2024 software, was used to identify, code, and categorize the most relevant text fragments. Each researcher carried out an analysis individually, and later, a comparison analysis was carried out in different meetings between the researchers and a third collaborator, which allowed the themes and categories to be adjusted, as well as improving the quality of the analysis, enhancing the results achieved. The established themes, categories, and codes were revised after a debate on them, reaching a consensus on the discrepancies seen.

Although the final interviews analyzed did not continue to add new information to the research, it was considered that a point of data saturation was not reached as all the interviews analyzed provided some information of value to this research.

In addition, for the age variable, the mean and standard deviation were calculated.

## 3. Results

### 3.1. Participants’ Profile

The main participants in this study were 15 informal caregivers of individuals with Alzheimer’s disease (58 ± 8.32). Table 2 shows the characteristics of the participants. The sample included 10 female caregivers (54.3 ± 5.6) and 5 male caregivers (65.4 ± 8.32). The participants were mainly related to the patients as sons or daughters, with sufficient income to provide care. Most of the participants had higher education or secondary education, and in relation to the marital status of the participants, they were married.

### 3.2. The Results of the Analysis

Following the thematic analysis of the interviews, 3 categories were found (Table 3), revealing the perception of the participants in different areas of the intervention, the application of the knowledge acquired by the participants, and the proposals for improvement for future interventions.


**Category 1. Benefits perceived by the participants**


Figure 1 shows 4 subcategories obtained: personal relationship, coping with illness situations, psychoemotional environment, and physical perception.


*
Personal relationship
*


Participants reported changes in the way they related to their family and close environment after the intervention.

“*You have to work hard in your head and be well so that you are not overcome by the situation so that it can’t get the better of you. It is true that it creates a lot of anxiety for the patient when the person who is with them is nervous or tense, so the calmer I am when I am with my mother, the calmer I will be when I am with her*” (Participant 6).

On the other hand, caregivers perceived improved social skills.

“*I have been working on social skills for some time now, but I think that the program has given me an important advance*” (Participant 2).

In addition, during the analysis, it was appreciated that the relationship created between the participants of the intervention had been a very important pillar for them.

“*We have been involved, we have supported each other, we have opened up to each other, we have supported each other when someone was weak*” (Participant 3).

Being in the same situation, they created new bonds between them, becoming part of their support network.

“*Seeing that there are other people who are going through situations very similar or the same as yours, makes me feel more supported in the sense that they can advise you, and above all, that they understand you*” (Participant 4).

“*To be in the same situation as other people, to see that you are not alone, to meet ordinary people, that people outside don’t understand and don’t know the burden that goes with it… and sometimes there are situations where you can’t, and coming to the intervention and seeing that I am not alone has unburdened me*” (Participant 15).


*
Coping with illness situations
*


The carers’ perception of the personal situation created by Alzheimer’s disease in their relative changed after their participation in the program.

“*I approach it from a different perspective. I approach it with hope and seeing more probabilities, and perhaps, seeing challenges not as problems but as opportunities*” (Participant 3).

In order for this situation not to overwhelm the informal carers, the attitude towards the situation had to be calm and patient.

“*At first, I was overwhelmed… until I realized that it was the illness, it wasn’t her… and through the tools we have, depending on whether she became stubborn, or repeated things a lot, or threw me, for example, I told myself… but, don’t you see? That helped me to understand and optimize my resources, achieving more patience above all*” (Participant 15).

In addition, the participants reported that they have acquired knowledge in the different areas taught in the program, and it has enabled them to act in a more confident way with their relatives.

“*I started to understand that it was easier to have knowledge and to know how you have to react. So, having that information made me feel confident and at the same time helped my mother*” (Participant 5).

Carers not only perceived changes in coping with the illness on a personal level, but they also saw changes in the way they deal with and/or interact with their family member.

“*Now I know that every time she says it is her first time, so I have been taught to be more patient and more empathetic with her and… to deal with it better*” (Participant 6).

For example, by reducing the aggressiveness, nervousness, and moodiness they sometimes showed towards the sick person.

“*Before, I could get a bit more irritable with some things that the relative did and now I have seen that these are things that the illness entails, and I try to be different*” (Participant 7).

“*It is a change above all in my attitude towards my relative, towards my environment… it has been good for controlling a little bit the episodes of aggression and nerves that often occur*” (Participant 8).


*
Psychoemotional changes
*


From a psychoemotional point of view, participants reported major changes, on the one hand, in the way they have begun to manage their emotions.

“*The program has brought me a lot of joy, because my personal situation at the moment was not particularly happy and taking part in the program has brought me an added joy, which I had not expected*” (Participant 3).

They also reported a reduction in the pressure and guilt they may feel in their day-to-day lives.

“*to say… we are caregivers, we are not superheroes, so remove a little bit of the guilt of if one day you can’t or one day you don’t arrive or one day you don’t know or one day you don’t give more because you don’t give more*” (Participant 3).

On the other hand, caregivers reported changes in their self-care. They started to give more importance and priority to themselves.

“*To value yourself and love yourself more, if you are well, the people around you will be well, and that is very important*” (Participant 9).

“*Give priority to you, don’t forget that if you fail as the caregiver, everything else fails, so give importance to you and your rest time*” (Participant 3).

Regardless of what other people may think, they started to do activities that they would not have done before their participation in the program. They have shown changes in their perception of the situation and have given more importance to their mental well-being and self-worth.

“*I think more about me. I don’t think about what people think, I just think about what I or can be good for me. For example, last week I went to a concert. A year ago, I wouldn’t have done it thinking that I would leave my husband at home, and now, I don’t, I looked for a person with whom he would be fine, and I did it, and a year ago, I wouldn’t have done it*” (Participant 10).

“*Before I had very low self-esteem, with a lot of anxiety, and learning to do relaxation exercises, increasing my self-esteem, valuing myself, because I didn’t value myself… it has been very good for me. Since the program started until now, I’ve changed 90%… no to say 100%*” (Participant 1).


*
Physical changes
*


Participants reported that the physical exercise program fostered a positive increase in their well-being.

“*It has brought me happiness. You really discover that you can do simple exercises which are not expensive to do, and also corrected and supervised, it is wonderful*” (Participant 11).

They reported an increase in the frequency of daily physical activity and, thus, the benefits of physical activity not only on a psychological level but also on a physical level, with strength being one of the variables in which they perceived the most changes.

“*I have been encouraged to exercise every day, I see it in a different way… it has been very good for me and I’m very happy about it*” (Participant 12).

“*Because, If the caregiver is not strong, if the caregiver does not take care of him/herself, he/she cannot continue to take care*” (Participant 5).

Having increased their strength, together with having learned how to carry out care techniques, has helped them greatly in their work as carers, being something fundamental for them, for the prevention of possible injuries in both the carer and the patient.

“*Because before I might have bent down in one way and… then I saw that it hurt, for example. You can see that I hadn’t done it properly, so no physics has helped me a lot to… for the strength… for the way to… I mean, to lift him… to lay him down… and all of that has helped me a lot*” (Participant 13).


**Category 2. Applicability of knowledge**


This second category was also found inductively after the interviews. It refers to the way in which participants apply and use the knowledge acquired during the program. Figure 2 illustrates the obtained subcategories, mobilization techniques, care and nutrition.


*
Care
*


To a greater extent, the interviewees referred to what they have learned in the area of health education. They reported having learned a lot of care techniques.

“*In the care classes, I had no idea about more than half of the things, and now I have the knowledge, which I will use when it comes*” (Participant 12).

Moreover, not only the caregivers attending the program benefited from learning from the program, as some interviewees reported that the people around them (or their immediate environment) have become more aware of the situation, learning different caregiving techniques as well.

“*I especially liked the topic of oral hygiene, as it was a topic I had not thought about, and since the course, we are more insistent on this topic at home. The truth is that it was very good*” (Participant 9).


*
Nutrition
*


Participants indicated the importance of their family members having good eating habits. Therefore, during the interviews, they revealed the personal satisfaction they felt in learning about nutrition and diet and thereby being able to provide their family members with better eating habits, directly impacting their mood.

“*At mealtimes, I am calmer… I feed him in a different way, if possible, without so much impatience after what I have learned here*” (Participant 4).


*
Mobilization Techniques
*


With the increase in physical activity and general knowledge of patient care, participants conveyed how all of this had greatly improved their ability to mobilize their family members in different situations.

“*It is noticeable, that there is more information and you learn more. For example, one of the techniques I use to lift my father I learned here on the subject of personal hygiene care*” (Participant 5).

“*Everything has been positive. Physically I was very neglected, and I had to support my mother a lot with her treatment and I didn’t have the right information to do things properly*” (Participant 12).


**
Category 3. Improvement Proposals
**


This category was found inductively after conducting the interviews and subsequent analysis. A subcategory was reported, as reflected in Figure 3, which shows the content distribution.


*
Content distribution
*


Unifying the weekly program hours or even increasing the number of weekly hours in a single session were some proposals made, always with the aim of having more personal use time per week.

“*I would have liked to unify the sessions of each area on a weekly basis, instead of doing two hours a week, or, to do the double time to make it compatible with our family and work life*” (Participant 14).

They also suggested altering the order of some sessions, considering that some theoretical aspects would be more suitable to be taught at the beginning of the program, all with the aim of better understanding the objectives of the practical activities.

“*We receive explanations about the benefits of physical exercise at the end of the program. It would be better to receive these explanations at the beginning of the program. In the beginning we received wonderful explanations about the physical exercises we were doing, but no explanations of why and what for, and what happens to oneself*” (Participant 2).

Lastly, participants expressed the need to increase the number of hours and importance of physical activity sessions, with the aim of increasing the frequency of physical exercise.

“*I would change that instead of doing the physical activity session every three weeks, I would do it once a week. In the health education psychology weeks, instead of doing one hour twice a week, I would do one and a half hours and the other physical exercise session*” (Participant 1).

## 4. Discussion

The main objective of this qualitative study was to learn about and explore the perceived benefits in terms of physical, caregiving knowledge and social support and psychological-emotional levels among informal caregivers of individuals with Alzheimer’s disease who participated in an interdisciplinary program. After conducting a thematic analysis of participants’ experiences in the program through semi-structured interviews, three categories emerged: perceived benefits by the participants, the applicability of acquired knowledge, and improvement proposals for future programs.

In general, participants highlighted how rewarding and beneficial it was for them to participate in a program of this nature. The negative effects on the health of caregivers of individuals with dementias such as Alzheimer’s disease have been documented. High levels of psychological and emotional burden [29], distress [30], vulnerability [31], and stress are some of the harmful effects on caregiver health. However, several studies have demonstrated how interventions targeting caregivers, focusing on psycho-educational aspects and family support, could reduce caregiver burden [30] and even that longer-duration interventions were more successful than short or single-session interventions [32].

The socio-demographic data of our study are in line with those seen in other studies. The majority of caregivers are female, with an average age between 53 and 56 years [29,33,34], married [29], and are typically related to the patient as daughter or wife [29].

On the other hand, the participants stated during the interviews how the knowledge acquired in each of the areas had become an indispensable tool for coping with the situation. In this sense, the results analyzed indicate how the participants perceived benefits in the way they relate socially, how they cope with the situation created by the disease, as well as finding benefits on a psychoemotional and physical level.

It is true that all participants reported physical and emotional improvement; however, these results are in line with other studies, which show associations between physical and emotional problems derived from caring for a person with dementia [35], given that Alzheimer’s disease affects both the personal life of the caregiver and the family [29]. On an emotional level, participants reported improved self-care, which has been directly associated with emotional well-being and overall health [36]. Furthermore, caregiver self-care prioritization may be influenced, according to Pope et al. [36], by personal characteristics, such as commitment to their well-being, ability to cope with challenges, or optimism. Therefore, due to the interdisciplinary nature of the program, physical activity may have had a positive influence on increasing self-care [37], as well as awareness of the importance of the caregiver in the whole caregiving process.

Regarding the personal relationships of the participants, they experienced changes within their immediate environment, as well as with the internal group created by the intervention. Participants expressed the number of activities and personal experiences they had stopped doing to care for the patient, something that has already been evidenced in other studies showing how informal caregivers tend to sacrifice time with friends and family, thereby reducing social contact and support [38,39]. However, finding people in the same situation, as well as the new relationship created with their close environment (family and close friends), is associated with increased self-care and better emotional management, leading to a decrease in burden or stress as they find tools and resources to cope with the situation [38], observing a direct positive relationship between social support and psychological well-being [40]. In this sense, the program became a psychological and social support group where participants felt they somehow shared the same burden and situations.

The new way of facing the situations was expressed by all participants, having acquired new knowledge and tools for it. They expressed it in two different ways: firstly, in how they faced the situation with the patient, and secondly, in their perception of the disease. Coping with a family member’s dementia illness creates higher levels of distress and burden for caregivers [6,13]. However, it was observed how some participants saw the situation in a better light than others in the same situation, making coping with the situation very complex. The ability to manage stress can improve both the ability to cope with the illness and the regulation of emotions, ultimately affecting the quality of life of caregivers [41].

Regarding the second category deduced from the interviews, we found how participants apply the knowledge acquired during the intervention program. However, no quantitative or qualitative study was found that has analyzed or evaluated how Alzheimer’s caregivers have used or applied the knowledge acquired during an interdisciplinary intervention with psycho-educational, cognitive–behavioral, educational, and physical training. In our study, during the interviews, participants expressed having implemented techniques for moving the patient in their daily routine. For a long time, many participants expressed not knowing the correct technique and position to avoid injuries for both the caregiver and the patient, and they greatly appreciated having learned how to move their family members in a safer way. On the other hand, caregiving techniques were very useful for all participants, as well as knowing tools that facilitate feeding the patient. Empowering the participant to become a more informed and capable caregiver has been associated with a lower level of burden [42]. Therefore, implementing these tools in the patient’s care routine to cope with the situation could decrease stress and distress related to moving or feeding the patient, personal self-care, and emotion management, ultimately reducing the emotional and psychological burden on the caregiver.

Firstly, knowledge on patient care and, secondly, on the utilization of these tools to cope with the situation reduced stress related to moving or feeding the patient, personal self-care, and emotion management, derived from knowing a greater number of patient care techniques, ultimately reducing the emotion and psychological burden on the caregiver.

As a final point to address, the third category, proposals for improvement, was mainly related to increasing the hours of physical activity in general. The benefits of physical activity have been more than evident. Physical activity is essential for maintaining good health [43], and its benefits in cardiovascular health and muscular strength development [44], as well as in overall well-being [45], have been well documented. Participants repeatedly expressed the need to incorporate at least one hour of physical activity every week instead of having physical activity sessions every few weeks. Anxiety or depression [46,47], which are sometimes present in caregivers of Alzheimer’s patients, are some of the diseases associated with physical inactivity. Therefore, weekly guided physical activity could become an ideal and necessary tool for caregivers.

In this study, we encountered several limitations, such as the nature of the study or the sample size, which did not allow us to generalize the results. Future research should aim to analyze the potential effects of an interdisciplinary intervention with a greater number of teaching hours per week.

## 5. Conclusions

From what was expressed by the participants of the Integral Care interdisciplinary program, it was understood that participating in the sessions contributed to improving the ability to manage emotions, establish social relationships, and cope with the situations created by the disease.

Furthermore, considering the increase in knowledge about the disease, the application of care techniques and patient mobilization can lead to the improvement of the physical, mental, and psychological health of informal Alzheimer’s caregivers.

## Figures and Tables

**Figure 1 healthcare-12-01414-f001:**
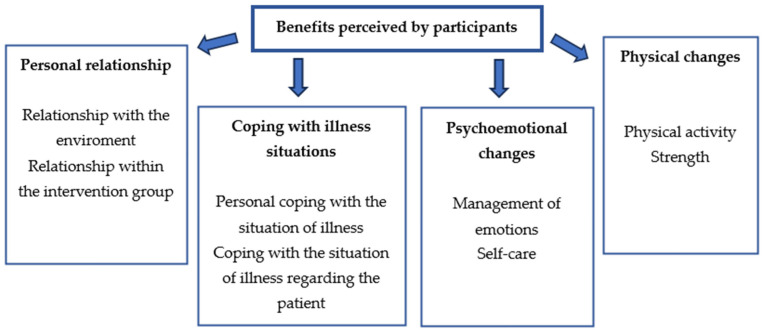
Theme and four main subcategories with the benefits perceived by the participants.

**Figure 2 healthcare-12-01414-f002:**
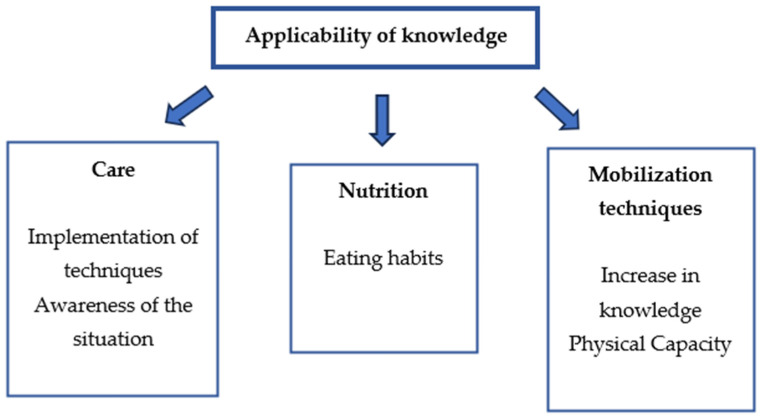
Theme and three main sub-categories with the applicability of knowledge.

**Figure 3 healthcare-12-01414-f003:**
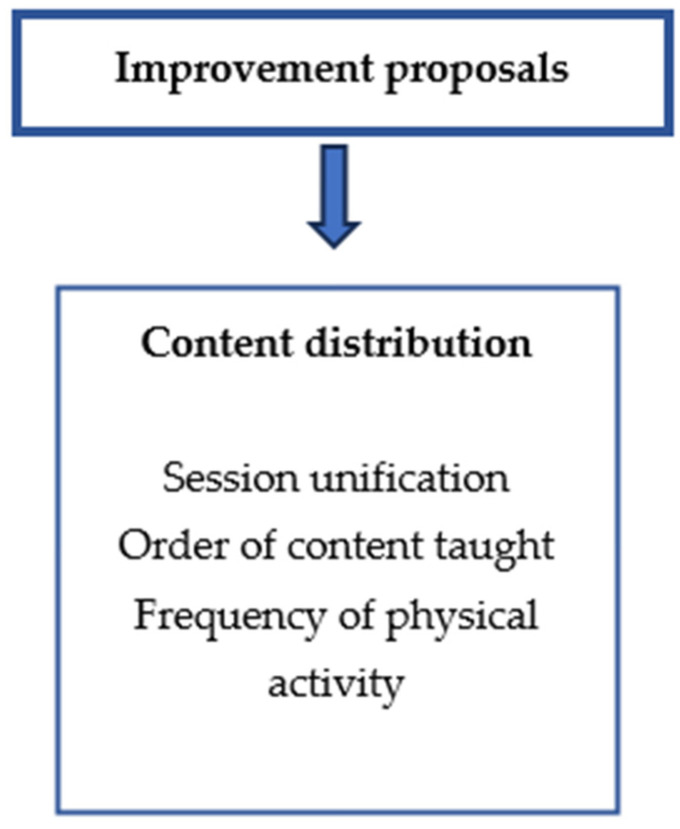
Theme and one subcategory with improvement proposals.

**Table 1 healthcare-12-01414-t001:** Areas and content.

Area of Intervention	Number of Sessions	Content/Module
Physical Activity	20	Flexibility
		Mobility
		Postural hygiene habits
		Upper body strengthening
		Lower body strengthening
Care and social support	20	Knowledge of the disease
		Care of the sick persons: mouth, skin, and digestive tract
		Nutrition
		Social and administrative support
Psychoemotional Education	20	Emotions
		Functions
		Professional balance
		Satisfaction and happiness
		Complementary alternatives

**Table 2 healthcare-12-01414-t002:** Sample of description.

Participant Number	Age Range	Gender	Degree of Relationship to the Patient	Sufficient Economic Income	Level of Education	Marital Status
1	51–55	Female	Son/daughter	Yes	Secondary School	Married
2	56–60	Male	Couple	Yes	Secondary School	Divorced
3	56–60	Male	Son/daughter	Yes	Higher Education	Single
4	66–70	Male	Couple	No	High School	Married
5	56–60	Female	Son/daughter	No	Secondary School	Married
6	56–60	Female	Son/daughter	Yes	Higher Education	Married
7	56–60	Female	Son/daughter	Yes	Higher Education	Single
8	61–65	Male	Son/daughter	Yes	High School	Single
9	46–50	Female	Son/daughter	Yes	Higher Education	Married
10	61–65	Female	Couple	No	Secondary School	Married
11	56–60	Female	Couple	No	High School	Single
12	51–55	Female	Son/daughter	No	Higher Education	Married
13	71–75	Male	Couple	Yes	Elementary Education (Reading and Writing Skills)	Married
14	46–50	Female	Son/daughter	Yes	Higher Education	Married
15	51–55	Female	Son/daughter	Yes	High School	Married

**Table 3 healthcare-12-01414-t003:** Categories and subcategories.

Category	Subcategory
Benefits perceived by participants	Personal relationship
	Coping with illness situations
	Psychoemotional domain
	Physical perception
Applicability of knowledge	Care
	Nutrition
	Mobilization techniques
Proposals for improvement	Content distribution

## Data Availability

The datasets used during the current study are available from the corresponding author on reasonable request.

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
