# Peer review of "Benefits Perceived by Caregivers of Patients with Alzheimer’s Disease on Physical and Emotional Health in an Interdisciplinary Program: A Qualitative Study"

_healthcare, 2024, doi:10.3390/healthcare12141414_

Round 1

Reviewer 1 Report

Comments and Suggestions for Authors

Dear authors,

I have enjoyed reading your study. It has valuable points and gives a support model for caregivers. My decision is to accept with minor revisions.

Comments:

Title – replace people with persons or patients

Affiliations – I am not sure the names of the research groups are accepted affiliations, department would be better.

Abstract: at the end of the abstract p1, l 31 add “of caregivers”.

Introduction:/

p.2. l 72 - Aim: word psychoemotional replace with emotional

-          2nd and 3rd paragraph start with “It is estimated” please revise

-          Replace “in the relative with dementia” with  “person with dementia”

-          p 2, l 64 “on the other hand” – there is no on the one hand.

Methods:

-          2.3. intervention the terms in the text (titles of intervention) do not correspond to the terms in table 2

-          Throughout the text there is a misunderstanding with the names of interventions.The names of the interventions do not correspond to their description. Cognitive-behavioural and health education is maybe “Care and social support” and Psychoemotional is “Mental health”

-          P.4. l 123 – not have pathologies is an exclusion criteria, please write it as such

-          Explain what data were you collecting (age, sex, etc)…

-          Add a statistical analysis section and explain how did you calculate data

-          where did the study took place and when?

-          Table 1 is not your own and therefore you can remove it or place it in the supplement.

-          Check the spelling in table 2

-          Figure titles go below the figures

-          Who gave the ethical approval ? And do you have written informed consent?

Results:

-          P.5, l.163-164 what are the values in brackets representing  - age? Meand+/- sd , explain

-          As the sample is small please calculate median age and 25-75 percentile ranges

-          3.1. please describe the sample and explain the table in more details - Degree of relationship to the patient, Sufficient economic income, Level of education and Marital status are not explained

-          The figures could be more informative if presented as tables. 
There is no info about the funding, ethics, etc at the end of the article. 

Discussion:

-          I don’t believe the gender data corresponds to other studies, caregivers are 90% female and yet you have 1/3 of male caregivers. You have to explain this.

Author Response

Thank you very much for your comments, they have undoubtedly contributed to improve the quality of the manuscript. The following are the responses to your comments.

REVIEWER 1

  1. Title: replacing people with people or patients

Authors' response: We appreciate your comment. We have changed the title as proposed.

  1. Affiliations: not sure that research group names are accepted affiliations, department would be better.

Authors' response: Thank you for your comment, your doubt is logical, but in our articles we always indicate our research group and there has been no problem.

  1. Abstract: at the end of abstract p1, l 31 add "of caregivers".

Authors' response: We agree with your contribution. We have added "of caregivers" at the end of the abstract to make it clearer whose benefits.

Introduction:/

  1. 2. l 72 - Objective: replace the word psychoemotional with emotional.
  2. The 2nd and 3rd paragraphs begin with "Esteemed" please revise.
  3. Replace " in the family member with dementia" with "person with dementia".
  4. p 2, l 64 "on the other hand" - there is no on the one hand.

               Authors' response: Thank you for your input. We have changed psychoemotional to emotional; revised the introduction of paragraphs 2 and 3; made the change from "family member" to "person" and; revised "on the other hand". All this has contributed to improve the introduction of the manuscript.

Methods:

  1. 3. intervention the terms in the text (intervention titles) do not correspond to the terms in Table 1.

                Authors' response: we have completed the table with the names of the interventions. We thank them for their contribution.

  1. Throughout the text there is a misunderstanding with the names of the interventions. The names of the interventions do not correspond to their description. Cognitive-behavioral and health education is perhaps "Social Care and Support" and Psychoemotional is "Mental Health".

Authors' response: We fully understand your comment. We have unified the terminology used in the intervention areas.

  1. 4. l 123 - no pathology is an exclusion criterion, please write it as such.

Authors' response: The criterion of "no pathology" has been left exclusively, so it does not lead to confusion.

  1. Explain what data you were collecting (age, sex, etc)...

Authors' response: Thank you for your comment, we have included this data in the description.

  1. Add a statistical analysis section and explain how you calculated the data.

Authors' response: We appreciate your input, however, we consider that the procedure for the analysis of the interviews has already been described in section 3.1. We must emphasize that no statistical analysis of the characteristics of the participants has been performed, they have simply been described.

  1. Where was the study conducted and when?

Authors' response: We have incorporated the date and where the study was conducted in section 2.2.

  1. Table 1 is not yours and therefore you can remove it or place it in the supplement.

Authors' response: Thank you for your appreciation, we place the table as a supplement.

  1. Check the spelling in table 2.

Authors' response: We have checked the spelling of the table.

  1. The figure captions go below the figures.

Authors' response: Thank you very much for your comment. We have changed the figure titles below the figures.

  1. Who gave ethical clearance and do you have written informed consent?

Authors' response: It is detailed in the Institutional Review Board Statement: Institutional Review Board Statement: This research was approved by the Bioethics and Biosafety Committee of the University of Extremadura (approval number 129/2020).

Results:

  1. 5, l.163-164 what do the values in parentheses represent - age? Mean+/- sd, explain

Authors' response: We have added that they represent the values in parentheses (mean+/-standard deviation) in section 2.5. 

  1. Since the sample is small, calculate the mean age and percentile ranks between 25 and 75.

Authors' response: The mean age of the participants is given in section 3.1.

  1. 1. Please describe the sample and explain the table in more detail - No explanation of the degree of relationship to the patient, Sufficient income, Level of education and Marital status.

Authors' response: Thank you for your input. You have described the sample and explained the variables in Table 2 in more detail.

  1. The figures could be more informative if they were presented in tabular form.

Authors' response: The categories and subcategories obtained from the qualitative analysis have been presented in tabular form (Table 3).

  1. There is no information on financing, ethics, etc. at the end of the article.

Authors' response: Funding is described after the conclusions section. Funding: This research is funded by the "Consejería de Economía, Ciencia y Agenda Digital de la Junta de Extremadura" and the European Social Fund, grant number #IB20175. The sponsors play no role in the design of the study, the decision to publish, or the preparation of the manuscript.

Discussion:

  1. I don't think the gender data corresponds to other studies, caregivers are 90% female and yet you have 1/3 male caregivers. You need to explain this.

Authors' response: Although in our study the percentage of women is less than 90%, female participants represent the majority of the sample in our study, which is why it is indicated in the discussion section. If you feel that we should add a more specific paragraph discussing these results, please let us know and we will make the change.

Reviewer 2 Report

Comments and Suggestions for Authors

Dear authors.

Thank you for the opportunity.

The article should be published because it’s necessary having more qualitative studies about “caregiving experience”. Despite the discussion should be improved, I approve the publication after mandatory reviews detailed below.

Review

1.      Review some of the “subjective” expression or provide objective data about it: “Alzheimer's disease (AD), the most common type of dementia” (L35)

2.      Improve the sentences to reduce repetitions: “It is estimated that” (L38) and “It is estimated that” (L41)

3.      Confirm the references: “Psycho-educational programs [7, 8]” (L56) and “Cognitive-behavioral and educational programs [9, 10]” (L58). References 7 and 8 are not about Psycho-educational programs, as well as the references 9 and 10. All references are about experiments. The references 12, 13, 14 and 15 are about educational programs.

4.      Confirm the references: “such as regular physical training [16, 17]” (L65). These references are about drugs and economy of pharmacies.

5.      Confirm the references “this sense, didactic exercise interventions have been developed for people with AD and their caregivers [21, 22]. (L67-68). The references are about “the Validity of the EQ-5D-5L” scale.

6.      Add more references of “qualitative studies” because they are the main approach of the article: “So far, several studies have evaluated from a qualitative perspective the process of adaption to the disease by family caregivers and the positive and negative experiences of caregiving [23, 24]” (L69-71)

7.      Review some of the “subjective” expression or provide objective data about it: “no study has evaluated the perceived benefits for caregivers of people with AD after participating in a program with psycho-educational, cognitive-behavioral, educational, and physical training interventions” (L71-73); “However, no quantitative or qualitative study was found that has analyzed or evaluated how Alzheimer’s caregivers have used or applied the knowledge acquired during an intervention.” (L484-486). Please consult: some examples of studies with a qualitative approach that can be improved. https://pubmed.ncbi.nlm.nih.gov/32757354/ , https://pubmed.ncbi.nlm.nih.gov/34401371/, https://pubmed.ncbi.nlm.nih.gov/32151166/, https://pubmed.ncbi.nlm.nih.gov/35239470/                                                                                                                                                                                                                                                                                                                                                                                                                                                                                                                                                                                                                                                                                                                                                                                                                                                                                                                                                                                                                                       

8.      Transfer the sentence to the section 2.1: “Approaching perception using qualitative methods can provide a more complete picture of program benefits in this population as psychosocial aspects, which are not easily measured, expressed, and interpreted from a quantitative perspective, help to gain insight and understanding of more complex emotional responses that quantitative methodology cannot reveal [26]. For this reason, a qualitative methodology is essential to provide a comprehensive understanding of the caregiver’s perspective.” (L75-80). This sentence is important to explain the methodology (design and procedures), and the use of the “qualitative approach”.

9.      Suggestion a new section’s organization for Section 2. I suggest: 2.1. Study design and procedure; 2.2. Intervention; 2.3. Sample and recruitment; 2.4. Interviews; 2.5. Analysis.

10.  In section 3.2., I suggest including more empirical references by subcategory, and resume them in a final table in an Annex. It’s important to analyse different “perspectives” (empirical data) by subcategory.

Comments on the Quality of English Language

-

Author Response

Thank you very much for your comments, they have undoubtedly contributed to improve the quality of the manuscript. The following are the responses to your comments.

REVIEWER 2

  1. Review some of the "subjective" expression or provide objective data on it: "Alzheimer's disease (AD), the most common type of dementia" (L35).

Authors' response: We understand your concern, but based on references 1 and 2 Alzheimer's disease is the most common cause of dementia. 

  1. Improve sentences to reduce repetition: "It is estimated that" (L38) and "It is estimated that" (L41).

Authors' response: Thank you for your comment. The introduction to paragraphs 2 and 3 has been revised.

  1. Confirm references: "Psychoeducational programs [7, 8]" (L56) and "Cognitive-behavioral and educational programs [9, 10]" (L58). References 7 and 8 do not deal with psychoeducational programs, as well as references 9 and 10. References 12, 13, 14 and 15 are about educational programs.

Authors' response: Thank you very much for your appreciation. We had an error in the bibliographic references and thanks to your comment, it has been solved.

  1. Confirm references: "such as regular physical training [16, 17]" (L65). These references are about drugs and pharmacy economics.

Authors' response: Thank you for your comment. An error has been detected in these references and the correct references have been included.

  1. Confirm references "in this regard, didactic exercise interventions have been developed for people with AD and their caregivers [21, 22]." (L67-68). The references are on "the validity of the EQ-5D-5L scale".

Authors' response: We thank them for their contribution. These references have been changed to the appropriate ones.

  1. Add more references to "qualitative studies" because they are the main focus of the article: "So far, several studies have evaluated from a qualitative perspective the process of adaptation to illness by family caregivers and the positive and negative experiences of caregiving [23, 24]" (L69-71).

Authors' response: Some possible references have been added as a comment to be added if the reviewer considers it necessary.

  1. Revise some of the "subjective" expressions or provide objective data: "no study has evaluated the benefits perceived by caregivers of people with AD after participating in a program with psychoeducational, cognitive-behavioral, educational, and physical training interventions" (L71-73); "However, no quantitative or qualitative study has been found that has analyzed or evaluated how caregivers of people with Alzheimer's disease have used or applied the knowledge acquired during an intervention" (L484-486). See: some examples of studies with a qualitative approach that can be improved. https://pubmed.ncbi.nlm.nih.gov/32757354/ , https://pubmed.ncbi.nlm.nih.gov/34401371/, https://pubmed.ncbi.nlm.nih.gov/32151166/, https://pubmed.ncbi.nlm.nih.gov/35239470/

Authors' response: We appreciate your comment. The text has been modified to avoid "subjective" expressions.       

  1. Transfer sentence to section 2.1: "Addressing perception using qualitative methods can provide a more complete picture of program benefits in this population, as psychosocial aspects, which are not easily measured, expressed, and interpreted from a quantitative perspective, help to gain. knowledge and understanding of more complex emotional responses that quantitative methodology cannot reveal [26]. For this reason, a qualitative methodology is critical to providing a comprehensive understanding of the caregiver's perspective." (L75-80). This sentence is important to explain the methodology (design and procedures) and the use of the "qualitative approach".

Authors' response: We appreciate your comment. We have included a sentence in part 2.1 to provide complete information on the qualitative methodology. We consider that the indicated sentence is important in the introduction for the justification of the methodology to be carried out.

  1. Suggested organization of a new section for Section 2. I suggest: 2.1. Study design and procedure; 2.2. Intervention; 2.3. Sample and recruitment; 2.4. Interviews; 2.5. Analysis.

Authors' response: We have made the organizational change in Section 2.

  1. In section 3.2., I suggest including more empirical references by subcategory and summarizing them in a final table in an Annex. It is important to analyze different "perspectives" (empirical data) per subcategory.

Authors' response: Thank you for your input. Figures 1, 2 and 3 address empirical references by subcatergories, indicating a summary in each of them of the insights obtained.
